# Recent Advances in Stroke Genetics—Unraveling the Complexity of Cerebral Infarction: A Brief Review

**DOI:** 10.3390/genes16010059

**Published:** 2025-01-06

**Authors:** Takeshi Yoshimoto, Hiroshi Yamagami, Yuji Matsumaru

**Affiliations:** 1Department of Stroke and Cerebrovascular Diseases, University of Tsukuba Hospital, Tsukuba 305-8576, Japan; yujimatsumaru@md.tsukuba.ac.jp; 2Division of Stroke Prevention and Treatment, Institute of Medicine, University of Tsukuba, Tsukuba 305-8575, Japan; yamagami.hiroshi@md.tsukuba.ac.jp; 3Department of Neurosurgery, Institute of Medicine, University of Tsukuba, Tsukuba 305-8575, Japan

**Keywords:** stroke genetics, polygenic risk score, *RNF213*-related vasculopathy, epigenetics, pharmacogenomics

## Abstract

Background/Objectives: Recent advances in stroke genetics have substantially enhanced our understanding of the complex genetic architecture underlying cerebral infarction and other stroke subtypes. As knowledge in this field expands, healthcare providers must remain informed about these latest developments. This review aims to provide a comprehensive overview of recent advances in stroke genetics, with a focus on cerebral infarction, and discuss their potential impact on patient care and future research directions. Methods: We reviewed recent literature about advances in stroke genetics, focusing on cerebral infarction, and discussed their potential impact on patient care and future research directions. Key developments include the identification of monogenic stroke syndromes, such as cerebral autosomal dominant arteriopathy with subcortical infarcts and leukoencephalopathy, and cerebral autosomal recessive arteriopathy with subcortical infarcts and leukoencephalopathy caused by mutations in the *NOTCH3* and *HTRA1* genes, respectively. In addition, the role of *RNF213* in moyamoya disease and other cerebrovascular disorders, particularly in East Asian populations, has been elucidated. The development of polygenic risk scores for assessing genetic predisposition to stroke has demonstrated the potential to improve risk prediction beyond traditional factors. Genetic studies have also elucidated the distinct genetic architecture of stroke subtypes, including large artery atherosclerosis, small vessel disease, and cardioembolic stroke. Furthermore, the investigation of epigenetic modifications influencing stroke risk and its outcomes has revealed new research avenues, while advancements in pharmacogenomics highlight the potential for personalized stroke treatment based on individual genetic profiles. Conclusions: These genetic discoveries have important clinical implications, including improved risk stratification, targeted prevention strategies, and the development of novel therapeutic approaches.

## 1. Introduction

Stroke remains a leading cause of death and disability worldwide, with a considerable global burden. In 2021, 93.8 million stroke survivors, 11.9 million new stroke cases, and 7.3 million stroke-related deaths globally were reported. Ischemic strokes accounted for 65.3% of all new strokes [1]. In Japan, the age and sex-adjusted incidence rate of first-ever ischemic stroke was estimated at 91.3 per 100,000 person-years, highlighting its substantial impact on public health [2]. Recently, considerable efforts have been made to understand the genetic underpinnings of stroke, particularly cerebral infarction [3,4,5,6]. Advances in genomic technologies, including genome-wide association studies (GWASs) and next-generation sequencing, have revolutionized the identification of genetic variants associated with stroke susceptibility and its outcomes [7,8]. These discoveries have enhanced our understanding of stroke pathophysiology and opened new avenues for personalized risk assessment, prevention strategies, and targeted therapies [1]. While stroke is generally considered a multifactorial disorder influenced by both environmental and genetic factors, its genetic contribution becomes particularly evident in young adults without traditional risk factors. The incidence of stroke in younger populations has been increasing, with > 60% of strokes occurring in individuals aged < 70 years and 16% in those aged < 50 years [9]. This trend underscores the importance of investigating genetic predispositions, especially in young adults with cryptogenic stroke. Moreover, the identification of monogenic stroke syndromes and exploration of polygenic risk scores have provided valuable insights into the complex genetic architecture of stroke [10]. The genetic landscape of stroke is vast and complex, encompassing rare monogenic disorders and polygenic variation. Recent studies have elucidated the roles of specific genes and genetic variants in stroke pathogenesis, including those involved in vascular integrity, thrombosis, and inflammation [11].

One area of particular interest is the emerging field of *RNF213*-related vasculopathy [12,13], which has been implicated in various cerebrovascular disorders, including the moyamoya disease (MMD) and intracranial arterial stenosis [14,15,16]. The identification of genetic risk factors for stroke has substantial clinical implications, including improved risk stratification, targeted prevention strategies, and the development of novel therapeutic approaches. As our understanding of stroke genetics increases, healthcare providers must remain informed about the latest developments in this rapidly advancing field. To this end, this review aims to provide a comprehensive overview of recent advances in stroke genetics, focusing on cerebral infarction, and discuss their potential impact on patient care and future research directions [17,18]; it focuses primarily on the intricate interplay between genetic factors and the risk of cerebral infarction.

## 2. Methods and the Genetic Architecture of Stroke

This brief review summarizes the current knowledge of stroke genetics based on a literature search of the PubMed and Web of Science databases between 2018 and 2023. We specifically examined studies addressing genetic variants, genome-wide association, and monogenic disorders associated with stroke.

The genetic architecture of stroke is complex and multifaceted, encompassing both rare monogenic disorders and polygenic variations. Monogenic etiologies account for 1% to 5% of all ischemic strokes [19,20], and etiology-based studies have provided valuable insights into the molecular mechanisms underlying cerebrovascular disease. Cerebral autosomal dominant arteriopathy with subcortical infarcts and leukoencephalopathy (CADASIL), caused by mutations in *NOTCH3*, is characterized by recurrent strokes, migraine with aura, and cognitive decline and follows an autosomal dominant inheritance pattern [21,22]. Mutations in the *HTRA1* gene cause cerebral autosomal recessive arteriopathy with subcortical infarcts and leukoencephalopathy, which presents with clinical features similar to those of CADASIL; however, unlike CADASIL, it follows an autosomal recessive inheritance pattern [23]. Other monogenic disorders, such as the Marfan syndrome [24,25], fibromuscular dysplasia [26], and MMD [12,13,27], are associated with nonatherosclerotic vasculopathy and an elevated risk of stroke.

Most patients with stroke have a polygenic basis, with multiple genetic variants that contribute to disease risk. The heritability of ischemic stroke is approximately 37.9%, varying by subtype as follows: 40.3% for large vessel disease, 32.6% for cardioembolic stroke, and 16.1% for small vessel disease [4]. GWASs have identified numerous loci associated with stroke risk, such as *HDAC9* gene variants associated with large artery atherosclerotic stroke and *PITX2* and *ZFHX3* gene variants linked to cardioembolic stroke [28]. The GIGASTROKE initiative, a large-scale GWAS meta-analysis involving 110,182 patients with stroke and 1,503,898 controls across five ancestries, identified 89 independent loci associated with stroke risk, 61 of which were novel [6]. Cross-ancestry studies have enhanced our understanding of stroke genetics across different populations, with 87% and 60% of the primary and secondary stroke risk loci, respectively, replicated across ancestries. Genetic evidence has highlighted potential drug targets, including *F11*, *KLKB1*, *PROC*, *GP1BA*, *LAMC2*, and *VCAM1*, as well as investigational drugs targeting *F11* and *PROC* [6]. Integrative polygenic scores, combining cross-ancestry- and ancestry-specific stroke GWASs with vascular risk factor GWASs, have shown strong predictive power for ischemic stroke across European, East Asian, and African ancestries [29]. Next-generation sequencing technologies, such as whole-exome and whole-genome sequencing, have enabled the identification of rare variants with potentially large effects on stroke risk [30]. These techniques have been particularly useful in uncovering novel monogenic causes of stroke and elucidating the genetic basis of familial stroke syndromes [31]. The rapid advancement in stroke genetics has substantial clinical implications, including improved risk stratification, personalized therapy, and the identification of new therapeutic targets [32]. Several other monogenic disorders have been identified as important causes of stroke. Fabry disease, an X-linked lysosomal storage disorder caused by mutations in the GLA gene, can lead to cerebrovascular complications, including stroke, particularly in young adults, accounting for 1% to 4% of cryptogenic strokes in younger patients [33,34,35]. *COL4A1*-related small vessel disease, caused by mutations in *COL4A1*, manifests with ischemic stroke and intracerebral hemorrhage, often accompanied by characteristic findings such as basal ganglia microbleeds and periventricular white matter changes [36,37]. Homocystinuria, resulting from mutations in the CBS gene, increases stroke risk through various mechanisms, including arterial and venous thrombosis, and should be considered when treating juvenile patients with ischemic stroke of unknown etiology [38,39]. In addition, mitochondrial disorders such as mitochondrial encephalopathy, lactic acidosis, and stroke-like episodes (MELAS) syndrome, caused by mutations in mitochondrial DNA, particularly the MT-TL1 gene encoding tRNA(Leu(UUR)), present with stroke-like episodes. MELAS is characterized by a slowly progressive spread of lesions and focal hyperemia [40,41]. These conditions further emphasize the diverse genetic landscape of stroke and the importance of considering its genetic etiologies, particularly in young patients or those with unusual presentations.

Advancing knowledge in stroke genetics would facilitate the development of more precise and effective stroke prevention and treatment strategies. The monogenic and polygenic causes of ischemic stroke are listed in Table 1 and Table 2.

## 3. *RNF213*-Related Vasculopathy: A Unifying Concept

The role of *RNF213* in cerebrovascular disorders has been recently discovered in stroke genetics. *RNF213* encodes a protein with both ubiquitin ligase and ATPase activities. Variants of this gene have been strongly associated with MMD, a rare cerebrovascular disorder characterized by a progressive stenosis of intracranial arteries [12,50,51,52,53]. In particular, the *p.R4810K* variant of *RNF213* has been identified as a major susceptibility factor for MMD in East Asian populations [52,53]. Recent studies have expanded the scope of *RNF213*-related vasculopathy beyond MMD [12,13,54]. *RNF213* variants have also been associated with other cerebrovascular phenotypes, including intracranial arterial stenosis and non-moyamoya intracranial major artery stenosis/occlusion [16]. These findings suggest that *RNF213* plays a broad role in cerebrovascular health and may contribute to stroke risk through multiple mechanisms. The exact mechanisms by which *RNF213* variants lead to vascular abnormalities and increased stroke risk remain unclear. However, recent studies have suggested that *RNF213* is involved in angiogenesis, vascular remodeling, and inflammatory responses [52,55,56,57,58]. Understanding these mechanisms may lead to the identification of novel therapeutic targets for stroke prevention and treatment.

Differentiating between adult MMD and intracranial atherosclerotic disease poses considerable challenges, especially when the vessels in MMD are underdeveloped or radiologically unidentifiable [59]. Consequently, patients with early or mild MMD who present with ischemic stroke may be misclassified as having large artery atherosclerosis [60] based on the Acute Stroke Treatment criteria (Trial of ORG 1017212) [61]. Studies have shown that the *pR4810k* variant of *RNF213* is associated with intracranial large artery stenosis or occlusion (ICASO), even in cases that do not satisfy the diagnostic criteria for MMD [14,15]. A study by Miyawaki et al. (2012) in Japan revealed that of the 41 patients with ICASO without signs of MMD, nine (22%) carried the *RNF213* variant [16]. Similar associations have been observed in the South Korean [62] and Chinese populations [14,63], although the strength of the association varies among East Asian countries.

Meta-analyses with large sample sizes and comprehensive datasets have demonstrated strong associations between the *pR4810k* variant of *RNF213* and MMD, ICASO, and moyamoya syndrome across different genetic models (such as dominant, recessive, and allelic models) [64]. These findings highlight the importance of this genetic variant in the pathogenesis of cerebrovascular disease. A landmark study investigating the *pR4810k* variant of *RNF213* in 46,958 Japanese individuals (17,752 patients with cerebral infarction and 29,206 healthy controls) revealed a high risk of cerebral infarction, particularly large artery atherosclerosis. The odds ratios were striking: 1.9 for total ischemic stroke and 3.6 for large artery atherosclerosis. Notably, this study also highlighted a sex disparity in risk, mirroring the pattern observed in MMD. Women carrying the variant had a higher risk of developing ischemic stroke than men (OR: 2.7 vs. 1.5) [27].

Accumulating evidence suggests a continuum between MMD and a subset of large artery atherosclerosis, now conceptualized as *RNF213*-related vasculopathy. [12,14] This condition is proposed to represent a mild form of MMD that predisposes carriers to hemodynamic compromise in intracranial atherosclerosis. [25,27,65,66] The spectrum of *RNF213*-related vasculopathy extends beyond MMD, moyamoya syndrome, and ICASO. Recent studies suggest the following:Hypoplasia of the extracranial cervical arteries [67]Vasospastic angina [68]Coronary artery disease [69,70]Systemic hypertension [18,71]Peripheral pulmonary hypertension/pulmonary arterial hypertension [72,73,74]Renal artery [75,76]Stenosis of the abdominal branches of the aorta [77,78]

This expanded view of *RNF213*-related vasculopathy underscores its role as a unifying concept in systemic vascular diseases, suggesting a common genetic basis for various vascular pathologies.

## 4. Polygenic Risk Scores and Stroke Prediction

The heritability of ischemic stroke has been well established in extensive genetic studies and was estimated at 37.9%, with marked variations across subtypes: 40.3% for large artery disease, 32.6% for cardioembolic stroke, and 16.1% for small vessel disease [28]. This varying heritability underscores the importance of subtype-specific genetic analyses in stroke research. Recent genetic studies have provided insights into the distinct genetic architectures of different stroke subtypes. Large artery atherosclerotic, small vessel, and cardioembolic strokes have been shown to have partially overlapping but distinct genetic risk profiles [79,80]. Genetic variants associated with atherosclerosis and lipid metabolism have been consistently identified in large artery atherosclerotic stroke [81]. A recent meta-analysis of GWAS data identified 22 novel loci associated with large artery stroke, including genes involved in blood pressure regulation and vascular smooth muscle cell function [82]. Small vessel strokes, including lacunar infarcts, are associated with genetic variants in blood pressure regulation and vascular integrity [83,84]. A study in 2021 identified novel loci associated with small vessel stroke, including genes involved in extracellular matrix homeostasis and endothelial function [83]. Cardioembolic stroke has been linked to genetic variants associated with atrial fibrillation and other cardiac abnormalities. Recent studies have uncovered novel genetic loci associated with cardioembolic stroke, including those involved in cardiac development and function [6]. Understanding the genetic basis of these stroke subtypes may provide a foundation for developing targeted preventive strategies and personalized treatment approaches.

Recent advances in cross-ancestry studies have further enhanced our understanding of stroke genetics. The GIGASTROKE initiative, which analyzed 110,182 patients with stroke across five ancestries, demonstrated that genetic risk scores were predictive of ischemic stroke independently of clinical risk factors in 52,600 clinical trial participants [6]. Integrative polygenic risk scores (iPRS) represent an advanced approach that combines multiple genetic components to enhance stroke predictions. The GIGASTROKE initiative demonstrated that iPRS, which integrates cross-ancestry- and ancestry-specific stroke GWASs with vascular risk factor GWASs, showed strong predictive power across European, East Asian, and African ancestries [6]. The Key Components of the Cross-Ancestry Stroke Risk Prediction Model are shown in Table 3.

Studies have shown that individuals with high polygenic risk have approximately a 2-fold higher risk of incident stroke than those with low polygenic risk [79]. However, it is noteworthy that in some populations, such as Chinese cohorts, current polygenic risk scores showed moderate association strengths, suggesting potential limitations in certain ethnic groups [85].

The integration of these scores with clinical risk factors has improved stroke risk prediction, although the magnitude of improvement varies across populations. Recent evidence suggests that combining clinical and polygenic risks can enhance stroke prediction, particularly in specific patient groups, such as those with atrial fibrillation [86].

## 5. Epigenetics and Stroke

Epigenetic modifications have emerged as important factors influencing stroke risk and outcomes [87,88]. Epigenetic changes such as DNA methylation and histone modifications can alter gene expression without altering the underlying DNA sequence. Environmental factors can influence these modifications and may provide a link between genetic predispositions and environmental risk factors for stroke. Recent studies have identified specific epigenetic markers associated with stroke risk and outcomes [89,90]. For example, a large-scale epigenome-wide association study identified differentially methylated regions associated with ischemic stroke, providing new insights into the epigenetic regulation of stroke-related genes. Epigenetic changes are also implicated in stroke recovery and rehabilitation [87,91]. Studies have shown that epigenetic modifications can influence neuroplasticity and functional recovery after stroke, suggesting potential targets for therapeutic interventions.

## 6. Pharmacogenomics and Stroke Treatment

Pharmacogenomics offers a promising approach to personalized stroke treatment based on an individual genetic profile. Genetic variations can influence drug metabolism and efficacy and associated adverse effects, making pharmacogenomic approaches particularly relevant for stroke management [92].

The role of *CYP2C19* polymorphisms in antiplatelet therapy responses has been extensively studied. Approximately 30% of the population carries at least one *CYP2C19* reduced-function allele, resulting in approximately one-third reduction in plasma exposure to the active metabolite [93]. *CYP2C19* loss-of-function alleles significantly affect platelet response to clopidogrel, with a notably high prevalence (10% to 25%) in Asian populations compared with 2% to 3% in Caucasians [94]. Recent studies have shown that *CYP2C19* genotyping allows clinicians to adjust treatment therapies, potentially improving outcomes in patients with ischemic stroke [95].

Beyond *CYP2C19*, other genetic variants also influence the effectiveness of stroke medication. The *CYP3A4*1G* variant has been identified as a protective factor against clopidogrel resistance, while polymorphisms in the *P2RY12* gene, which encodes the pharmacological target of clopidogrel, affect platelet reactivity [96]. In addition, variants in *APOA5* and *CETP* significantly influence the effectiveness of statin therapy in patients with ischemic stroke [96]. Recent studies have explored the potential of genotype-guided antiplatelet therapy to improve the outcomes of these patients [94,97,98,99].

Pharmacogenomics of anticoagulation therapy remains an area of active research. Well-established genetic variants affecting warfarin metabolism support the monitoring of warfarin dosing in some populations. Research is ongoing to identify the genetic factors that influence the efficacy and safety of newer anticoagulants [100].

## 7. Future Directions and Clinical Implications

As advancements in stroke genetics continue, several key areas for future research and clinical application have emerged. One priority is the integration of genetic information into clinical risk-prediction models. Although polygenic risk scores have shown promise, further research is required to optimize their performance and validate their utility in diverse populations. Another important area is the development of targeted therapies based on genetic insights. As we uncover the molecular mechanisms underlying genetic risk factors for stroke, new therapeutic targets may emerge. For instance, studies on *RNF213*-related vasculopathy hold the potential for developing treatments for MMD and other cerebrovascular disorders. The field of precision medicine in stroke care is poised to advance with genetic information-guided treatment decisions and preventive strategies. This could include the genotype-guided selection of antiplatelet or anticoagulation therapy and personalized risk reduction strategies based on the genetic risk profile of an individual.

## 8. Conclusions

Remarkable progress has been made in the field of stroke genetics in recent years, providing valuable insights into the complex genetic architecture of cerebral infarction and other stroke subtypes. Advancements, such as identifying monogenic stroke syndromes and developing polygenic risk scores, have enhanced our understanding of stroke pathophysiology and opened new avenues for personalized prevention and treatment strategies. As research continues to unravel the genetic complexities of stroke, integrating genetic information into clinical practice holds great promise for improving stroke prediction, prevention, and management. Future studies will undoubtedly refine our understanding of stroke genetics and translate these findings into tangible benefits for individuals at risk and patients with cerebrovascular disease.

## Figures and Tables

**Table 1 genes-16-00059-t001:** Monogenic causes of ischemic stroke.

Mechanism	Disease	Gene	Inheritance Pattern	Features
Small vessel disease	CADASIL [21,22]	*NOTCH3*	AD	Migraine, recurrent TIAs and lacunar infarcts, VCI, and hyperintensities in the white matter of the anterior temporal lobe.
	CARASAL [42]	*CTSA*	AD	Migraine, TIA, stroke, dysphagia, VCI, and REM sleep dysfunction
	PADMAL [43]	*COL4A1*	AD	Recurrent lacunar infarcts, pons predominance, and progressive cognitive impairment
	COL4A1-related small vessel disease [36,37]	*COL4A1*	AD	Hemorrhagic and ischemic stroke, cerebral small vessel disease, microbleeds, leukomalacia, congenital cataracts, glaucoma
	CARASIL [23]	*HTRA1*	AR	Autosomal recessive inheritance, similar to CADASIL
	Homocystinuria [38,39]	AR	CBS	Arterial and venous thrombosis, stroke, cognitive impairment, lens dislocation, skeletal abnormalities
	MELAS [40,41]	Maternal	MT-TL1 and others	Stroke-like episodes, encephalopathy, lactic acidosis, myopathy, hearing impairment, diabetes, cardiomyopathy
	Fabry disease [33,34,35]	*GAL*	X-linked recessive	Cerebrovascular complications, stroke (particularly in young adults), acroparesthesia, skin rashes, gastrointestinal issues, and kidney and heart problems
Large vessel disease	Marfan syndrome [24,25]	*FBN1*	AD	Dissection of the ascending aorta and marfanoid features
	Moyamoya disease [12,13,27]	*RNF213*	AD or AR	Terminal internal carotid artery steno-occlusive disease with “smoke puff” collaterals
	Sickle cell disease [44]	*HBB*	AR	Crises of pain, convulsions, myelopathy, and anemia

AD, autosomal dominant; AR, autosomal recessive; TIA, transient ischemic attack; VCI, vascular cognitive impairment; REM, rapid eye movement; CADASIL, cerebral autosomal dominant arteriopathy with subcortical infarcts and leukoencephalopathy; CARASIL, cerebral autosomal recessive arteriopathy with subcortical infarcts and leukoencephalopathy.

**Table 2 genes-16-00059-t002:** Polygenic causes of ischemic stroke.

Gene/Locus	Associated Stroke Subtype	Risk Allele Frequency	Effect Size (Odds Ratio)	Population Studied
*HDAC9* [45]	Large vessel stroke	0.20 [46]	1.42 [45]	European ancestry
*PITX2* [3]	Cardioembolic stroke			European ancestry
*ZFHX3* [3]	Cardioembolic stroke	0.19	0.73 [47]	European and Chinese Han populations
*RNF213* [12,13,27]	Ischemic stroke	0.052	2.50	European and Chinese Han populations
*F11* [6]	Any stroke, any ischemic stroke, cardioembolic stroke		1.07	European descent
*KLKB1* [48]	Ischemic stroke			European, East Asian, African, South Asian, Latin American
*PROC* [49]	Cardioembolic stroke	0.20	1.16	European ancestry

**Table 3 genes-16-00059-t003:** Key Components of the Cross-Ancestry Stroke Risk Prediction Model.

Component	Description	Predictive Value
Cross-ancestry GWAS	Incorporates genetic data from 110,182 stroke patients across five ancestries	Identifies 89 independent stroke risk loci [6]
Vascular Risk Factors	Includes genetic variants associated with blood pressure, type 2 diabetes, lipids, BMI, and atrial fibrillation	Enhances prediction accuracy [79]
Ancestry-specific Variants	Considers population-specific genetic architecture	Improves prediction across different ethnic groups [85]

## Data Availability

No new data were created or analyzed in this study. Data sharing is not applicable to this article.

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
