# Peer review of "Recent Advances in Stroke Genetics—Unraveling the Complexity of Cerebral Infarction: A Brief Review"

_genes, 2025, doi:10.3390/genes16010059_

Round 1

Reviewer 1 Report

Comments and Suggestions for Authors

Dear Authors,

Thank you for your article. The prevalence of cerebral infarction is rising globally, particularly among younger individuals. This highlights the urgent need for more studies exploring the genetic predisposition to such incidents.

Introduction:
You may consider including statistical data on the global prevalence of stroke, or prevalence specifically in Japan, to provide greater context and relevance to your study. Additionally, stroke is often described as a multifactorial disorder; however, in specific cases, such as in young individuals without traditional risk factors, genetic predisposition should be strongly suspected. This point could be emphasized further to strengthen your argument.

Materials and Methods:
This section appears to be missing from your work and would significantly enhance the rigor of your study. Consider including details on whether you followed PRISMA guidelines, the number of articles screened, and your inclusion and exclusion criteria. Providing this information will improve the transparency and reproducibility of your research.

Genetic Architecture of Stroke:
You could expand this section by discussing additional monogenic disorders associated with stroke, such as Fabry disease, COL4A1-related small vessel arteriopathy with hemorrhage, homocystinuria, and mitochondrial disorders. If you prefer not to delve into detailed descriptions of these conditions, you might consider expanding the table provided in your article to include these disorders.

Polygenic Risk Score:
This section could be enhanced by incorporating more information about the heritability of cerebral infarction. Adding such data would provide a more comprehensive understanding of the genetic factors at play.

Pharmacogenomics Section:
This section is currently too brief. You could elaborate further by discussing additional polymorphisms that influence the metabolism of drugs commonly used in the management of ischemic stroke. Expanding on this point will make this section more robust and clinically relevant.

Your article requires major revisions.

Author Response

  1. Responses to Reviewer #1

Thank you for your comments. The prevalence of cerebral infarction is rising globally, particularly among younger individuals. This highlights the urgent need for more studies exploring the genetic predisposition to such incidents.

A-1. Introduction:
You may consider including statistical data on the global prevalence of stroke, or prevalence specifically in Japan, to provide greater context and relevance to your study. Additionally, stroke is often described as a multifactorial disorder; however, in specific cases, such as in young individuals without traditional risk factors, genetic predisposition should be strongly suspected. This point could be emphasized further to strengthen your argument.

Response to Comment A-1: Thank you for your insightful suggestion. The Introduction has been revised to incorporate comprehensive statistical data on global stroke prevalence, as well as specific data for Japan. In addition, we have strengthened the emphasis on genetic predisposition in young patients with stroke without traditional risk factors. The revised text now includes the following.

  1. Global statistics of stroke burden (93.8 million survivors, 11.9 million new cases, and 7.3 million deaths in 2021)
  2. Japan-specific data on stroke incidence
  3. Enhanced discussion on stroke in younger populations
  4. Stronger emphasis on genetic predisposition in young adults with cryptogenic strokes

These revisions are reflected in the first and second paragraphs of the Introduction. I believe these additions substantially strengthen the context and relevance of our review.

A-2. Materials and Methods:
This section appears to be missing from your work and will significantly enhance the rigor of your study. Consider including details on whether you followed PRISMA guidelines, the number of articles screened, and your inclusion and exclusion criteria. Providing this information will improve the transparency and reproducibility of your research.

Response to Comment A-2: We thank the reviewer for this valuable suggestion. We would like to clarify that this manuscript was conceived as a brief review to provide a concise overview of recent advances in stroke genetics rather than a systematic review or meta-analysis. In light of this, we have included a brief methodological statement at the beginning of Section 2, outlining our approach to literature selection. We acknowledge that a full PRISMA-compliant systematic review would necessitate a more comprehensive methodology; however, it is beyond the scope of this brief review. The revised manuscript now includes a concise description of our literature search approach, which we believe is appropriate for the format and scope of this brief review, while ensuring transparency regarding our methodology.

A-3. Genetic Architecture of Stroke
This section can be expanded by discussing additional monogenic disorders associated with stroke, such as Fabry disease, COL4A1-related small-vessel arteriopathy with hemorrhage, homocystinuria, and mitochondrial disorders. If you prefer not to delve into the detailed descriptions of these conditions, consider expanding the table provided in your article to include these disorders.

Response to Comment A-3: Thank you for this suggestion. In response, we have expanded the section on the genetic architecture of stroke by incorporating a new paragraph that comprehensively discusses additional monogenic disorders, including Fabry disease and its X-linked inheritance pattern, COL4A1-related small vessel disease, homocystinuria and its vascular complications, MELAS syndrome, and other mitochondrial disorders. Furthermore, Table 1 has been updated to include these conditions, their inheritance patterns, causative genes, and key clinical features. These revisions provide a more comprehensive overview of monogenic disorders associated with stroke risk while maintaining the concise nature of our brief review. We believe that these changes considerably strengthen the manuscript and provide readers with a broader understanding of the genetic landscape of stroke.

A-4. Polygenic Risk Score:
This section can be enhanced by incorporating more information about the heritability of cerebral infarction. Adding such data would provide a more comprehensive understanding of the genetic factors at play.

Response to Comment A-4: We thank the reviewer for the valuable suggestion. We have enhanced this section by incorporating specific heritability data for overall ischemic stroke (37.9%) and its subtypes. These changes provide a more comprehensive understanding of stroke heritability and its implications in risk prediction.

A-5. Pharmacogenomics Section:
This section is currently too brief. You could elaborate further by discussing additional polymorphisms that influence the metabolism of drugs commonly used in the management of ischemic stroke. Expanding on this point will make this section more robust and clinically relevant.

Response to Comment A-5: We have substantially expanded this section by including detailed information about CYP2C19 polymorphisms and their prevalence across different populations. In addition, we discussed other genetic variants, such as CYP3A4*1G and P2RY12, provided information about genetic influences on statin therapy response, and incorporated relevant citations from recent literature. These additions provide a more comprehensive overview of how genetic variations influence the effectiveness of stroke medication, thereby enhancing the clinical relevance of the section.

Reviewer 2 Report

Comments and Suggestions for Authors

This is a well-written concise review dealing with recent advances in elucidation of the genetics of stroke. The introduction is clear and concise and gives an appropriate context for the article. The authors provide a suitable timely summary of the genetic architecture of stroke and then  focus in more detail on RNF123-related vasculopathy. This is followed by shorter and more superficial sections on epigenetics and pharmacogenomics; these sections seem somewhat limited given the multitude of data published in this area. There is no information on miRNAs and lncRNAs. The conclusions are supported by the text of the article. Overall, this is a good but incomplete article that briefly summarizes the knowledge in the field of stroke genetics.

Minor comments:

Authors describe within part 2, genetic architecture of stroke, the monogenic disorders and polygenic based stroke. However, there is a table for monogenic causes of ischemic stroke but there is not a table supporting the polygenic area. I would recommend including a summary for that too. 

Table 1 should include references for each line item.

Line 121, MDD should be changed for MMD.

Reference 62 seems not correct; it should be updated.

Also, authors should expand on integrative polygenic risk scores (line 85, line 208), perhaps with a table to complement the text.

Author Response

  1. Responses to Reviewer #2

This is a well-written concise review dealing with recent advances in elucidation of the genetics of stroke. The introduction is clear and concise and gives an appropriate context for the article. The authors provide a suitable timely summary of the genetic architecture of stroke and then  focus in more detail on RNF123-related vasculopathy. This is followed by shorter and more superficial sections on epigenetics and pharmacogenomics; these sections seem somewhat limited given the multitude of data published in this area. There is no information on miRNAs and lncRNAs. The conclusions are supported by the text of the article. Overall, this is a good but incomplete article that briefly summarizes the knowledge in the field of stroke genetics.

Minor comments:

The authors describe the genetic architecture of stroke, monogenic disorders, and polygenic-based stroke in part 2. However, there is a table for monogenic causes of ischemic stroke, but there is no table supporting the polygenic area. I would recommend including a summary for that, too. 

B-1. Table 1 should include references for each line item.

Response to Comment B-1: We appreciate the reviewer’s valuable suggestion. In response, we have revised Table 1 and added Table 2

B-2. Line 121, MDD should be changed for MMD.

Response to Comment B-2: We have made the necessary corrections.

B-3. Reference 62 seems not correct; it should be updated.

Response to Comment B-3: We have made the necessary corrections.

B-4. Also, authors should expand on integrative polygenic risk scores (line 85, line 208), perhaps with a table to complement the text.

Response to Comment B-4: We have included a table (Table 3), as suggested, to expand on integrative polygenic risk scores.
